# Genome-Wide Identification and Evolution Analysis of the Gibberellin Oxidase Gene Family in Six Gramineae Crops

**DOI:** 10.3390/genes13050863

**Published:** 2022-05-12

**Authors:** Chenhao Zhang, Xin Nie, Weilong Kong, Xiaoxiao Deng, Tong Sun, Xuhui Liu, Yangsheng Li

**Affiliations:** 1State Key Laboratory of Hybrid Rice, College of Life Sciences, Wuhan University, Wuhan 430072, China; zch_nx@whu.edu.cn (C.Z.); weilong.kong@whu.edu.cn (W.K.); 2017102040003@whu.edu.cn (X.D.); stong_fx@whu.edu.cn (T.S.); 2020202040079@whu.edu.cn (X.L.); 2State Key Laboratory of Agricultural Microbiology, College of Life Science and Technology, Huazhong Agricultural University, Wuhan 430070, China; nxnx_nx@163.com; 3Shenzhen Branch, Genome Analysis Laboratory of the Ministry of Agriculture, Agricultural Genomics Institute at Shenzhen, Chinese Academy of Agricultural Sciences, Shenzhen 518120, China

**Keywords:** Gramineae, gibberellin oxidase, orthogroups, abiotic stress, expression patterns

## Abstract

The plant hormones gibberellins (GAs) regulate plant growth and development and are closely related to the yield of cash crops. The GA oxidases (GAoxs), including the GA2ox, GA3ox, and GA20ox subfamilies, play pivotal roles in GAs’ biosynthesis and metabolism, but their classification and evolutionary pattern in Gramineae crops remain unclear. We thus conducted a comparative genomic study of *GAox* genes in six Gramineae representative crops, namely, *Setaria italica* (Si), *Zea mays* (Zm), *Sorghum bicolor* (Sb), *Hordeum vulgare* (Hv), *Brachypodium distachyon* (Bd), and *Oryza sativa* (Os). A total of 105 *GAox* genes were identified in these six crop genomes, belonging to the *C*_19_*-GA2ox, C*_20_*-GA2ox, GA3ox,* and *GA20ox* subfamilies. Based on orthogroup (OG) analysis, *GAox* genes were divided into nine OGs and the number of *GAox* genes in each of the OGs was similar among all tested crops, which indicated that *GAox* genes may have completed their family differentiations before the species differentiations of the tested species. The motif composition of GAox proteins showed that motifs 1, 2, 4, and 5, forming the 2OG-FeII_Oxy domain, were conserved in all identified GAox protein sequences, while motifs 11, 14, and 15 existed specifically in the GA20ox, C_19_-GA2ox, and C_20_-GA2ox protein sequences. Subsequently, the results of gene duplication events suggested that *GAox* genes mainly expanded in the form of WGD/SD and underwent purification selection and that maize had more *GAox* genes than other species due to its recent duplication events. The cis-acting elements analysis indicated that *GAox* genes may respond to growth and development, stress, hormones, and light signals. Moreover, the expression profiles of rice and maize showed that *GAox* genes were predominantly expressed in the panicles of the above two plants and the expression of several *GAox* genes was significantly induced by salt or cold stresses. In conclusion, our results provided further insight into *GAox* genes’ evolutionary differences among six representative Gramineae and highlighted *GAox* genes that may play a role in abiotic stress.

## 1. Introduction

Gibberellins (GAs) form a large family of diterpenoid compounds and are involved in plants’ development and growth in various aspects, including seed germination [1], stem elongation [2], leaf expansion [3], flowering [4,5], root growth, and fruit development [6,7,8]. The first discovery of gibberellin was in the pathogenic fungus *Gibberella fujikuroi*, and the fungus-infected plants showed an excessive elongation phenotype; this phenomenon is known as ‘foolish-seedling’ [9]. To date, more than 130 GAs have been discovered in plants, bacteria, and fungi, of which only a few GAs harbor biological activity, namely, GA_1_, GA_3_, GA_4_, and GA_7_ [10]. The majority of GAs have no biological activity in plants, acting as precursors of bioactive substances or inactivated metabolites [11].

GA signaling pathways have been described in many plants, and the biosynthetic pathway of GA is catalyzed by seven key enzymes: ent-copalyl diphosphate synthase (CPS), ent-kaurene synthase (KS), ent-kaurene oxidase (KO), ent-kaurenoic acid oxidase (KAO), GA 13-oxidase (GA13ox), GA 20-oxidase (GA20ox), and GA 3-oxidase (GA3ox) [12]. The main mechanism involves the 2β-hydroxylation catabolism of active GA, which is catalyzed by GA 2-oxidase (GA2ox) [13]. In view of GA, homeostasis is essential for plants’ normal growth and development, and various mechanisms have been developed to regulate the level of bioactive GA in plants [14]. 

*GAox* genes are involved in the synthesis and degradation of gibberellin in plants [15]. The *GA2ox*, *GA3ox*, and *GA20ox* genes belong to the 2OG-Fe (II) oxygenase superfamily and encode multifunctional gibberellin oxidase [16]. GA20ox and GA3ox are two types of key enzymes in the last step of the gibberellin biosynthesis pathway and catalyze GA_12_ and GA_53_ to GA_1_ and GA_4_, respectively [17], while GA2oxs inactivate GAs and their precursors, maintaining the balance of GA activity in plants [18]. According to their preference for C_20_-GA or C_19_-GA substrates, *GA2oxs* are further classified into *C*_19_*-GA2oxs* and *C*_20_*-GA2oxs*, respectively. Phylogenetic analysis also supported *C*_19_*-GA2ox*s and *C*_20_*-GA2oxs* belonging to two subfamilies [18,19].

The *GAox* genes have been identified in several plant species. In *Arabidopsis*, 19 *GAox* genes (ten *GA2ox* genes, four *GA3ox* genes, and five *GA20ox* genes) have been identified [20,21]. There are 24 presumed *GAox* genes in the Grape (*Vitis vinifera* L.) genome, including 11 *GA2ox*, 6 *GA3ox,* and 7 *GA20ox* genes [22]. In addition, 13 *GAoxs* have been identified in *Liriodendron chinense* genomic data [23]. The members of *GAox* genes harbor multiple functions. For instance, both the knockout of *GA20oxs* or *GA3oxs* and the overexpression of *GA2ox* genes affect the biological activity of GAs in plants, resulting in plant dwarfing [24], with the Green Revolution gene *sd1* (*OsGA20ox2*) greatly contributing to increased grain yields [25]. Furthermore, *GAox* genes regulate GA levels in plants to maintain normal growth and development under abiotic stresses. For example, the overexpression of the *OsGA2ox-5* gene can significantly improve the resistance to salt stress in rice [26]. *AtGA20ox-2* was reported to play an important role in response to high salinity, and the inactivation of *AtGA20ox-2* leads to the elongation of mutant root length and a decrease in sodium ion content relative to wild-type root [27]. Moreover, transcript levels of *AtGA2ox-9* increase after cold treatment and the *AtGA2ox-9* loss-of-function mutants are more sensitive to the freezing temperatures [20]. Furthermore, *GAox* genes also respond to biotic stress, and the RNAi lines of *OsGA20ox-3* show enhanced resistance to rice blast and bacterial blight [28].

Multiple *GAox* genes are involved in domestication. The semi-dwarf phenotype has been widely selected as an important agronomic trait in the breeding process of modern crops. *SD1* (*OsGA20ox1*) was involved not only in modern breeding including the Green Revolution but also in the early domestication of rice. *SD1* was subject to artificial selection during rice evolution, and FNPs (functional nucleotide polymorphisms) were involved in the domestication of japonica rice, suggesting that ancient humans had used Green Revolution genes [29]. In addition, the copy number variations of *gibberellin 2-oxidase 8* genes reduced trailing growth and branch length during soybean domestication [30]. The phylogenetic analysis of *GAox* genes that have been shown to be involved in evolutionary transitions may be an important means of addressing questions about evolutionary history or processes.

Gramineae belong to the monocotyledonous flowering plants and contain a variety of vital cereal crops with high economic value and rich genetic resources. *GAox* genes have been identified previously in several Gramineae [31,32,33], but the accuracy of genome assembly has made a great leap in recent years. In this scenario, we identified some new *GAox* genes and explored their evolution in six Gramineae crops (*Oryza. sativa* (Os)*, Zea. mays* (Zm), *Hordeum vulgare* (Hv)*, Brachypodium distachyon* (Bd)*, Setaria italica* (Si)*,* and *Sorghum bicolor* (Sb)) at the genome-wide scale. We systematically analyzed the phylogenetic relationships, gene structures, motif compositions, orthogroups (OGs), chromosomal locations, duplication events, selective forces, and cis-elements in the promoters of *GAox* genes. In addition, the RNA-seq data were used to analyze *GAox* genes’ tissue expression patterns and their responses to cold or salt stress in rice and maize. Our results provided a basis for further study on the gene expansion, evolutionary patterns, and functional diversity of *GAox* gene families in Gramineae.

## 2. Materials and Methods

### 2.1. Plant Materials and Treatments 

Nipponbare (*Oryza. Sativa*) rice seeds were cultivated with Youshida medium in the 26 °C greenhouse of Wuhan University, the light/dark photoperiod was 16/8 h, and relative humidity was 60% [34]. At three-leaf stage, some of the seedlings were transferred to Youshida medium with 125 mmol/L NaCl for salt treatment. The growth temperature of the other seedlings was changed to 4 °C for cold treatment. The roots were collected at 0, 3, and 24h after stress treatments, and three biological replicates for each treatment were conducted.

### 2.2. Identification of GAox Genes

The genome databases of Bd (v3.0), Hv (IBSC_V2), Si (v2.0), Sb (NCBIv3), Zm (B73_RefGen_v4), and Os (MSU 7.0) were downloaded from Ensembl Plants (http://plants.ensembl.org/index.html, accessed on 1 November 2020) or TIGRDatabase (http://rice.plantbiology.msu.edu, accessed on 1 November 2020). Seventeen OsGAox protein sequences from UniPort (www.uniprot.org, accessed on 5 December 2020) were used as query sequences for searching OsGAox protein sequences in protein datasets of Bd, Hv, Si, Sb, Zm, and Os using BLASTP v2.9.0 (E-value of e-5), respectively. In addition, the Hidden Markov Model (HMM) profile of 2OG-FeII_Oxy (PF03171) was obtained from the Pfam database (http://pfam.janelia.org, accessed on 6 December 2020).

A total of 17 OsGAox proteins as seed sequences were searched in protein databases by BLASTP v2.9.0 (E-value of e-5) and Hmmsearch v3.3.1 (with default parameters) methods, respectively [20,35], and the results of the two search methods were intersected. Subsequently, all putative GAox protein sequences were submitted to Pfam (http://pfam.xfam.org/search/sequence, accessed on 18 December 2020) and SMART (http://smart.embl-heidelberg.de, accessed on 18 December 2020) to detect key conserved domains [36,37], and the final candidate genes were determined after eliminating sequences with incomplete 2OG-FeII_Oxy (PF03171). 

### 2.3. Phylogenetic Tree Construction of GAox Genes

For phylogenetic analysis, the final candidate GAox proteins of each species were aligned by MUSCLE v3.8.1551 [38], trimAl v1.4 was used to filter gap and nonconservative sites [39], and then the phylogeny trees were generated by iqtree v2.0.3 with 1000 bootstrap replicates [40]. To ensure the accuracy of phylogenetic tree, the optimal model identified by iqtree was adopted to rebuild the tree by RaxML v0.9.0 [41]. Then, the Evolview online site (https://www.evolgenius.info/evolview/, accessed on 10 January 2021) was used to visualize the phylogenetic tree [42]. 

### 2.4. Homology Analysis of GAox Genes 

The lineal ortholog gene groups of *GAoxs* in six plants were analyzed by OrthoFinder v2.4.0 software [43]. An all-vs.-all BlastP search of all OsGAox protein sequences was performed by BlastP through diamond v0.9.24.125 software with default parameters as the input file for OrthoFinder software [44]. Finally, the candidate *GAox* genes were named according to their homology in rice [32]. In addition, DnaSP 5.0 (http://www.ub.edu/dnasp/, accessed on 25 January 2021) software was used to calculate the value of Tajima’s D for each OG [45].

To explore the expansion modes and differentiation times of *GAox* genes in six Gramineae, the duplication events were analyzed by MCScanX software [46]. First, we used Diamond software to perform BlastP among the *GAox* genes in each species [44], and then the ‘duplicate_gene_classifier’ script in the MCScanX software was adopted to identify the paralogous genes with the default parameters. The nonsynonymous/synonymous (Ka/Ks) substitution rates of the identified gene pairs were calculated by DnaSP 5.0 [45], and the divergence times were estimated by T = Ks/(2 × 9.1 × 10^−9^) × 10^−6^ million years ago (Mya) [47].

### 2.5. Distribution and Structure of GAox Genes

The chromosomal positions of *GAox* genes were obtained through GFF3 files of six species and visualized through Tbtools v1.089 [48]. All putative GAox protein sequences were submitted to the MEME Suite 5.3.3 (http://meme-suite.org/tools/meme, accessed on 12 February 2021) with the maximum number of motif sets at 15 and the optional width of motifs from 6 to 150 to check conservative motifs [49]. Both gene structure and conservative motifs were visualized by Tbtools [48].

### 2.6. Cis-Elements and Expression Analyses of GAox Genes

The 2kb upstream sequences of the transcription start sites of *GAox* genes were submitted to PLANTCARE (http://bioinformatics.psb.ugent.be/webtools/plantcare/html/, accessed on 23 February 2021) for cis-element prediction [50]. The *GAox* genes’ expression profile data of different tissues in maize and rice were obtained from MBKBASE website (http://www.mbkbase.org/rice, (accessed on 3 March 2021) and ZEAMAP (http://www.zeamap.com/, accessed on 3 March 2021), respectively [51,52]. The RNA-seq data of rice under cold stress (GSE57895) were obtained from NCBI (https://www.ncbi.nlm.nih.gov/gds, accessed on 24 March 2021), and the salt stress data were from our previous research [53]. The expression data of maize under abiotic stress were taken from ePlant (http://bar.utoronto.ca/eplant_maize/, accessed on 24 March 2021). The tissue expression levels of *GAox* genes were represented by Fragments Per Kilobase Million (FPKM) values, log2 (treament/CK) was calculated to represent *GAox* genes’ expression under stress treatment, and the heatmaps of gene expression were drawn with the R package (pheatmap).

### 2.7. RNA Extraction and qRT-PCR Analysis

Total RNA from the samples were extracted by TRIzol reagent. Purified total RNA were reversed to first-strand cDNA using HiScript III 1st Strand cDNA Synthesis Kit (Vazyme, Shanghai, China). The qTR-PCR analysis was conducted by 2 × SYBR Green qPCR Master Mix (US Everbright^®^ Inc., Suzhou, China) according to the manufacturer’s instructions. The qRT-PCR reactions were detected with CFX96 TouchTM Real-Time PCR Detection System (Bio-Rad, Hercules, CA, USA). The actin gene (UBI) was used as an internal reference control, and the primers in this experiment were designed by Primer Premier 5.0 (Appendix A). The relative expression level was calculated based on three biological replicates using 2^−^^△△CT^ method [35].

## 3. Results

### 3.1. Identification and OG Analysis of GAoxs in Six Gramineae

A total of 105 nonredundant GAox proteins were identified in six Gramineae crops: *O. sativa* (17), *Z. mays* (20), *H. vulgare* (17), *S. bicolor* (17), *S. italica* (15), and *B. distachyon* (19) (Appendix A, Figure 1). The number of *GAox* genes was similar in the six species mentioned above, with the largest number of *ZmGAoxs* and the smallest number of *SiGAoxs*. In order to explore the evolutionary model of *GAox* genes among six Gramineae, both iqtree and RaxML were used to construct a phylogenetic tree based on an edited multiple-sequence alignment (MSA) file (Appendix A), and the results of the two software were consistent, which proved the reliability of the phylogenetic tree (Appendix A). *GAox* genes could be clearly divided into four groups, as the *GA20ox, GA3ox*, *C*_19_*-GA2ox,* and *C*_20_*-GA2ox* subfamilies (Figure 1A) [54]. Subsequently, we identified nine orthologous groups using OrthoFinder (Appendix A), and *GA3ox* genes corresponded to OG-7, *GA20ox* genes were classified into OG-8,9, and the remaining six OGs all belonged to *GA2ox* genes (Figure 1A,B). In each OG group, the number of orthologous genes among the tested species was nearly equal. For example, each tested species contained two *GAox* genes belonging to OG-4 or OG-7 (Figure 1B). These results strongly indicated that the evolution of *GAox* families occurred before the separation of the major grass families. 

Then, we calculated Tajima’s D-values to evaluate the selection pressure of each OG. Tajima’s D-values of nine OGs ranged from −0.34 to −0.89 (Figure 1B), and this result emphasized that all OGs were subjected to varying degrees of purification selection. Among them, *GA3ox* genes (OG-7) had the highest Tajima’s D-value, and we speculated that owing to the function of *GA3ox* genes being indispensable to plants and that there were only two members in each species, it faced strong selection pressure. This part of the results implied that *GA3ox* genes were relatively conservative, while in *GA2ox* and *GA20ox* genes, gene loss/gain may have occurred in the course of the evolution of six Gramineae after the separation of the major Gramineae families.

### 3.2. Chromosome Localization, Gene Duplication Events, and Selective Forces Analysis

In order to explore the expansion patterns of paralogous genes, we separately executed the gene positions and duplication events of *GAox* genes in six species. *GAoxs* were unevenly distributed on chromosomes (Chrs) (Figure 2). For example, five and six *GAox* genes were located on chromosomes 1 and 5 in rice, respectively. However, the other chromosomes only harbored 0–2 genes (Figure 2A). Interestingly, we noticed that *GA20ox-1* genes from six species were commonly located on the edge of the chromosome, and the positions of *GA2ox-4* and *GA2ox-8* were in proximity in five plants except for Si. Thus, we speculated that the physical positions of some *GAox* genes were conservative during the Gramineae plant evolution process. 

Gene duplication events were some of the main drivers of genomics evolution [55]. Based on our results, gene duplication events were found in all tested species (Table 1) and also detected in the *GA2ox, GA3ox,* and *GA20ox* subfamilies, respectively. Their expansions were mainly through whole-genome duplication or segmental duplication events (WGD/SD), while tandem duplications (TD) only existed in Hv, Bd, and Si *(*Figure 2B,D,E). Notably, some paralogous pairs were simultaneously identified in several plants, such as *GA2ox-3. GA2ox-4* as well as *GA2ox-6* gene pairs existed in Os, Hv, Zm, Si, and Sb, and *GA2ox-9* gene pairs occurred in Os, Zm, Si, Bd, and Sb, implying that these paralogous genes had been duplicated prior to the ancestral divergence *(*Figure 2). 

Subsequently, the divergence times and selection pressures of paralogous gene pairs were evaluated by the Ka/Ks values. The results ranged from 0.19 to 0.87 (Table 1), meaning that all gene pairs were subjected to purification selection. Moreover, the divergence times of all duplicated gene pairs ranged from 2.09 to 66.43 Mya. The WGD/SD events were frequently noted in *GAox* genes’ early expansion (27.76–66.43 Mya), and the divergence time of gene pairs in OG-1 were the earliest, 39.18–66.4 Mya. During later evolution, *GAox* genes’ expansion was by two ways: WGD/SD and TD (4.24–26.84 Mya). Maize harbored the most *GAox* genes, which was due to the recent doubling of several pairs of genes, such as *ZmGA2ox-7a-ZmGA2ox-7b, ZmGA2ox-3a-ZmGA2ox-3b,* and *ZmGA20ox-2a-ZmGA20ox-2b*.

### 3.3. Gene Structure and Conserved Motif Analysis

The *GAox* gene structures were visualized based on the GFF3 files of six species (Figure 3B). The lengths of coding sequences and the numbers of exons in all tested *GAox* genes were various, of which most *GAox* genes consisted of 1 to 3 exons. However, even in the same OGs, some genes contained long introns, while others had only one exon, such as OG-1. Therefore, it was speculated that in *GAox* genes, sequence loss and exon fusion might occur during evolution. 

We executed the MEME online tool and searched 15 motifs in 105 GAox protein sequences (Appendix A). The composition and arrangement of motifs in four subgroups were similar (Figure 3A). The conserved domain of 2OG-FeII_Oxy consisted of motifs 1, 2, 4 and 5, which were identified in all GAox protein sequences (Figure 3A). It was worth noting that the tails of GAox protein sequences contained specific motifs that could distinguish different subfamilies. For example, motif 11, motif 14 and motif 15 existed only in *GA20ox, C*_19_*-GA2ox,* and *C*_20_*-GA2ox,* separately. Furthermore, motif 7 existed specifically in *the GA3ox* and *GA20ox* genes. GA3ox and GA20ox proteins were responsible for the synthesis of gibberellin, while GA2ox proteins degraded gibberellin [24], so motif 7 may have been related to the substrate recognition of *GAox* gene families. GA2ox proteins were classified as *C*_20_*-GA2ox* or *C*_19_*-GA2ox* genes based on their preference for C_20_-GA or C_19_-GA substrates [20], and their differences in conserved domains were reflected in motifs 8, 14, and 15. In general, our results revealed that *GAox* families formed unique conserved motifs in the process of differentiation.

### 3.4. Cis-Elements Analysis of GAoxs

We identified a total of 31 cis-elements distributed unevenly on the promoter region which were involved in growth and development, stress response, light response, and phytohormone (Figure 4A). Growth- and development-related cis-elements accounted for 8% (Figure 4B), involving endosperm development, circadian rhythm, meristem, and metabolism. Thirteen percent of the cis-elements were relevant in the stress response, which was associated with drought, low temperature, and anoxic specific inducibility. A total of 16.3% of cis-elements were predicted to respond to a light signal, such as G-box, GATA-motif, MRE elements, and so on. Moreover, phytohormone-related cis-elements (43%) included auxin responsiveness (AuxRR-core), gibberellin responsiveness (TATC-box, P-box, and GARE-motif), and MeJA-responsiveness (CGTCA-motif and TGACG-motif). Previous studies had also successively proved the various functions of *GAox* genes [56,57]. These results indicated that *GAox* genes may execute multiple biological functions and play a key role in plant growth, development, and stress.

### 3.5. Tissue-Specific Expression of GAox Genes 

We retrieved the expression data of rice and maize from the public database and our previous study (Figure 5A,B; Appendix A). *GAox* genes were widely expressed in rice or maize tissues. In rice, *OsGA2ox-10* and *OsGA20ox-1* were highly expressed in most tissues (Figure 5A). On the contrary, the expression levels of *OsGA2ox-2, OsGA2ox-4, OsGA2ox-5, OsGA2ox-6,* and *OsGA2ox-8* were low in most tissues (Figure 5A). The majority of *OsGAox* genes were predominantly expressed in rice panicles, indicating that *OsGAox* genes were vital in panicle development, which was consistent with previous research results [58]. Moreover, the expression profiles of paralogous gene pairs were various (Figure 5A). For example, *OsGA2ox-10* was expressed highly in most tissues, while its paralogous genes *OsGA2ox-7* were predominantly expressed in flowers and panicles, which revealed that these gene pairs underwent functional differentiation after duplication events. 

There were many similarities between the *GAox* genes’ expression patterns in rice and maize. In maize, *ZmGAox* genes were predominantly expressed in ear primordium, which was consistent with that in rice. This result showed that *GAox* genes played a key role in panicle development of both maize and rice. Interestingly, *GA2ox-10* and *GA20ox-1* were clustered together in rice or maize tissue expression profiles, and two pairs of genes showed similar expression patterns, indicating that their expression in Gramineae seemed to be very conservative and played similar biological functions. 

### 3.6. GAox Genes Responded to Cold or Salt Stress

In plants, *GAox* genes participated in plenty of physiological processes, especially in resisting abiotic stress, which played a significant role. Many researchers have investigated the performance of *GAox* genes under abiotic stress [20,59]. Under salt stress, there was little difference in the expression patterns of *GAoxs* between Chao2R (*indica*) and RPYgeng (*japonica*). For example, *OsGA20ox-2* and *OsGA20ox-3* were up-regulated and *OsGA2ox-3* and *OsGA2ox-*8 were down-regulated in both varieties (Figure 5C, Appendix A). For cold stress, we obtained the transcriptome data of *GAox* families from TNG67 (*japonica*) under low temperature for 3 h and 24 h (Figure 5C). Most members of *OsGAox* genes were sensitive to cold stress, and the expression of *OsGA2ox-3* altered significantly. Moreover, *OsGA3ox-1* responded differently to cold stress in various tissues, which was down-regulated in roots but up-regulated in shoots.

In our study, *ZmGAox* genes mostly responded to salt or cold stress by down-regulating their expression, while *ZmGA2ox-10* and *ZmGA2ox-6* were up-regulated under both stresses (Figure 5D). There were several *ZmGAox* genes which showed the opposite expression pattern under salt or cold stress, such as *ZmGA2ox-7a, ZmGA2ox-9, ZmGA2ox-11,* and *ZmGA20ox-1*. It was worth noting that the great differences appeared in the expression patterns of homologous gene pairs in rice and maize. The exception was that *ZmGA2ox-1a* and *ZmGA2ox-1b* clustered together in the maize stress expression profile, and they may have been functionally redundant in response to stress. In order to verify the correctness of the RNA-seq data, we tested the expression levels of *GAoxs* in the roots of japonica rice (Nipponbare) treated with cold and salt for 3h and 24h, respectively. Five representative *OsGAox* genes that were significantly differentially expressed under stress were selected to perform qRT-PCR, and the test results showed that these genes were significantly induced by stress (Figure 5E). This work was consistent with the analysis of the cis-elements of promoters. Most *GAoxs* were induced by cold or salt stress, and some *GAox* genes may have been functionally redundant.

## 4. Discussion

Bioactive gibberellins (GAs) are diterpene plant hormones that are biosynthesized through complex pathways and control diverse aspects of growth and development [10]. In addition, in the 1960s, the regulation of GAs on plant height had a significant influence on agriculture, and the cultivation of semi-dwarf wheat and rice contributed to a substantial increase in food production, which was called the Green Revolution [60]. Owing to the great significance of gibberellin oxidase in plant growth and development, we herein systematically identified 105 *GAox* genes in six Gramineous plant genomes. The number of *GAox* genes in each tested species was similar. The smallest genome (genome size: 490 Mbp) of *S. italica* contributed the fewest number of *GAox* genes. Although maize had the most *GAox* genes among six plants, it is obvious that *ZmGAox* genes did not double with their special genome-wide duplication [61]. Moreover, maize had the most *GAox* genes, and gene duplication events showed that they had experienced doubling recently, which meant that their functions may have been redundant, and our expression profile results also provided relevant proof, such as that *ZmGA2ox-1a* and *ZmGA2ox-1b* showed similar expressions in stress response.

The final stage of bioactive GAs synthesis is catalyzed by two *GAox* families, called the *GA20ox* and *GA3ox* genes [12,24]. Moreover, in the pathway of GA degradation, biologically active GAs or their immediate precursors are inactivated by two *GA2ox* families, including the *C*_19_*-GA2ox* and *C*_20_*-GA2ox* genes [10]. The above four *GAox* families all belong to the 2-ODDs superfamily and share an extremely conservative domain [62,63]. However, three families perform completely different functions [64,65]. In order to explore the sequence differences between them, motif identification was performed and revealed that some common motifs were shared by the four *GAox* subfamilies, while some unique motifs existed only in specific families. In our results, the *GAox* families (*GA3ox* and *GA20ox* genes) related to GA biosynthesis specifically contained motif 7 compared to the families (*GA2ox* genes) responsible for degradation, and motif 11 could further distinguish the *GA3ox* and *GA20ox* genes. The difference in motif composition between the *C*_19_*-GA2ox* and *C*_20_*-GA2ox* families depended on the existence of motif 14 (Figure 3A). In summary, the four *GAox* subfamilies all harbored unique motifs that could distinguish each other, which may have been the result of the differentiation of the 2-ODDs superfamily. Whether these motifs are related to substrate binding is worthy of further exploration.

In the previous study, Han and Zhu identified eight *GA20oxs*, while we only identified four *OsGA20ox* genes in our Blastp results by using *Arabidopsis*’ *GAox* genes [21]. We speculated that the previous incomplete genome, various parameters, or softs led to the different results. Liu et al. conducted functional analysis of *OsGA20ox7*, there was no significant change of GA content in the leaves of two *OsGA20ox7* mutants generated by CRISPR/Cas9, and they speculated the role for *OsGA20ox7* related to salicylic acid (SA) metabolism [66]. Another study found that *OsGA20ox-5* and *OsGA20ox-8* clustered in the *CsGA7ox* subfamily, which was a type of multifunctional enzyme with 7-oxidase and 3β- and 15 α-hydroxylase activity. This result indicated that *OsGA20ox5* and *OsGA20ox8* were homologs of *CsGA7ox1* and *CsGA7ox2* [67]. Furthermore, our conservative motif analysis also confirmed that *OsGA20ox1**–**4* were significantly different from *OsGA20ox5-8* (Appendix A). In general, *OsGA20ox5**–**8* may not belong to the *GA20ox* subfamily, and the functional study of *OsGA20ox5**–**8* will further confirm it.

Here, we confirmed that the evolution of *GAox* families took place before their ancestors divided from several aspects. First, orthogroups analysis classified *GAox* genes into nine OGs, and the number of *GAox* genes in each OG was similar in each tested species (Figure 1B), which indicated that these OGs were generated prior to six species’ divergence. In addition, the gene duplication event is one of the driving factors of species evolution [55]. We identified 33 pairs of homologous genes with duplication events, and the tested species commonly shared some of gene pairs (Table 1), showing that the duplication of these gene pairs also occurred before the differentiation of six plants. Furthermore, the *GA20ox-1* genes of all six species were located in the tails of chromosomes (Figure 2), indicating that they may have been located in the tails of chromosomes before the differentiation of the tested species. Thus, based on the above evidence, we speculated that the evolution of *GAox* families occurred before the ancestral separation. 

The results of cis-acting elements analysis showed *GAox* genes may respond to a variety of biological processes, including growth and development, stress response, light response, and phytohormone. Tissue expression profiles showed that *GAox* genes were highly expressed in both rice and maize, which indicated *GAox* genes may regulate plant panicle development. A previous study demonstrated that *OsGA20ox1* contributed to increasing grain number and yield by regulating cytokine activity in rice panicle meristems [58]. In addition, the expression data of *GAox* genes under cold or salt stress were retrieved and analyzed. The results showed that the majority of *GAox* genes responded to cold or salt stress and the expression of *OsGA2ox-3*, *OsGA20ox-2, ZmGA2ox-7a, ZmGA2ox-6, and ZmGA2ox-10* changed significantly. Previous studies had proved that the application of *GAox* genes can improve plant stress resistance [20,68,69]. For example, the ectopic expression of *GA2ox6* mutants can result in reduced plant height, increased yield, and improved abiotic and biotic stress tolerance in transgenic rice [68]. Therefore, future research on the function of *GAox* genes may provide valuable gene resources for stress-resistant breeding in Gramineae.

## 5. Conclusions

In this study, 105 gibberellin oxidase genes were identified in Gramineous crops, which can be subdivided into four subfamilies and nine OG groups. The similar number of GAox genes in the six species suggested that they may have evolved prior to the isola-tion of six Gramineous crop, and mainly expanded with WGD/SD expansion methods as well as undergone purification selection. The cis−acting elements analysis indicated that GAox genes may respond to growth and development, stress, hormones, and light signals. Moreover, the expression profiles of rice and maize showed that GAox genes were pre-dominantly expressed in the panicles of the above two plants and the expression of sever-al GAox genes was significantly induced by salt or cold stresses. In summary, this study provides a comprehensive insight into the evolution of GAox genes in six Gramineous crops.

## Figures and Tables

**Figure 1 genes-13-00863-f001:**
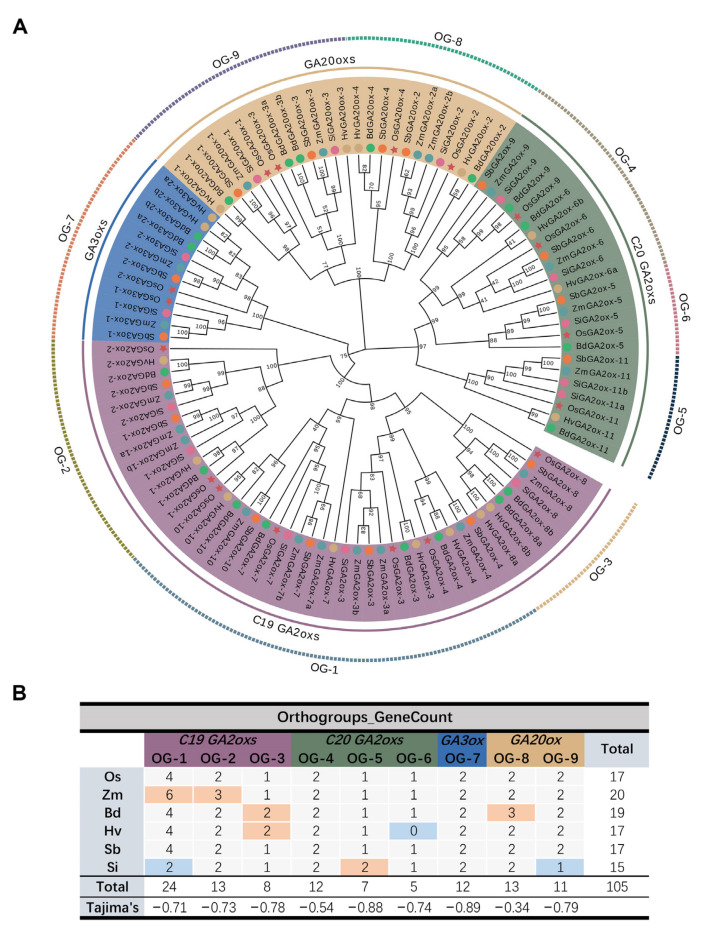
Identification and orthologous analysis of *GAox* genes. (**A**) A phylogenetic tree constructed by 105 protein MSA files. *GAox* genes of six species are labeled by various colors. The four colored backgrounds (purple, blue, yellow, and green) represent four subfamilies. Nine OGs are annotated by dotted lines with different colors. (**B**) Statistical analysis of orthologous genes among six plants and Tajima’s D-values of nine OGs. Orange and blue backgrounds represent expanded or lost species compared to rice in the same OG group, respectively.

**Figure 2 genes-13-00863-f002:**
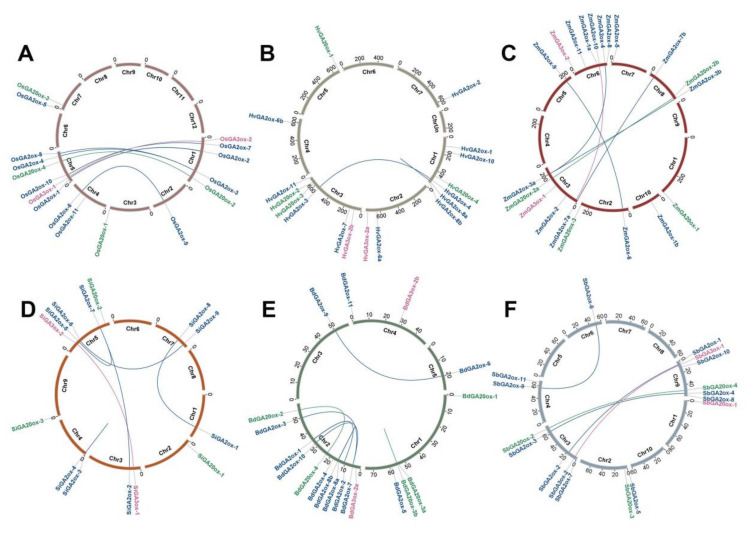
The chromosome location and duplication events of *GAox* genes in six species, namely, *O. sativa* (**A**), *H. vulgare* (**B**), *Z. mays* (**C**), *S. italica* (**D**), *B. distachyon* (**E**), and *S. bicolor* (**F**). The blue, purple, and green lines, respectively, represent the duplication events within the *GA2ox, GA3ox,* and *GA20ox* genes in each species.

**Figure 3 genes-13-00863-f003:**
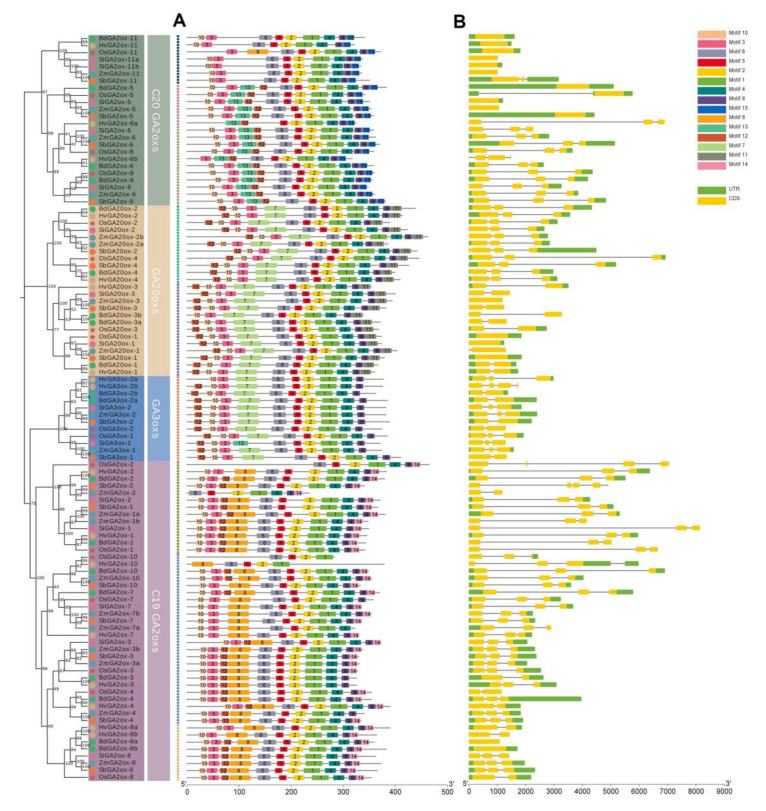
Phylogenetic tree, gene structure, and conserved motif analysis of *GAox* genes in six plants. (**A**) Conserved motif analysis, where the different colored boxes represent different motifs. (**B**) Gene structure analysis, where the green and yellow boxes represent UTR and exons, separately, and introns are displayed by gray lines.

**Figure 4 genes-13-00863-f004:**
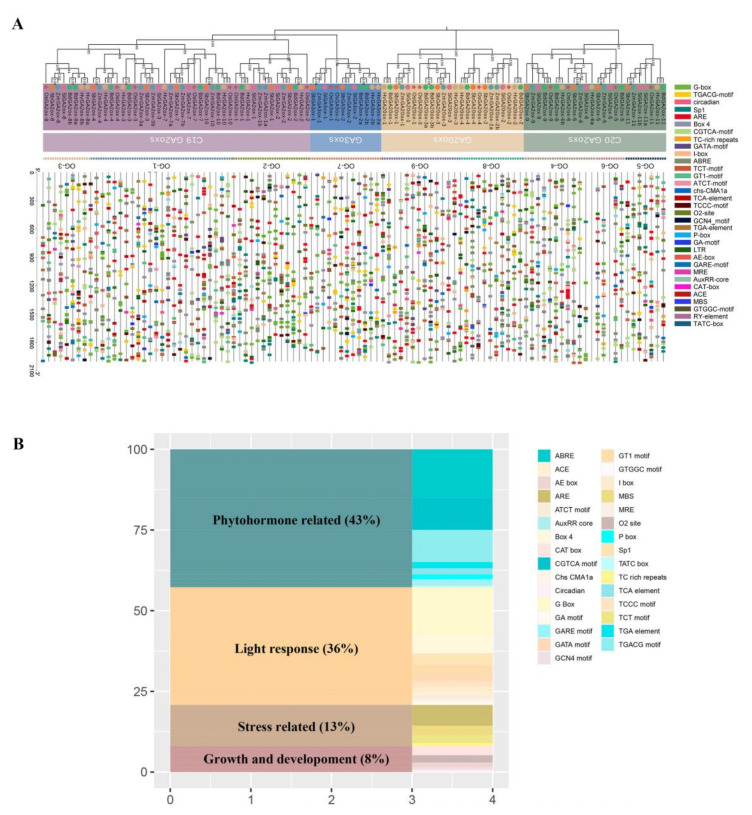
Analysis of cis-acting elements in *GAox* genes’ promoter regions. The legend describes the color of the cis-element. (**A**) Phylogenetic tree and the distribution of *GAox* genes’ cis-elements in six Gramineae. (**B**) The ratios of cis-elements classified by functions.

**Figure 5 genes-13-00863-f005:**
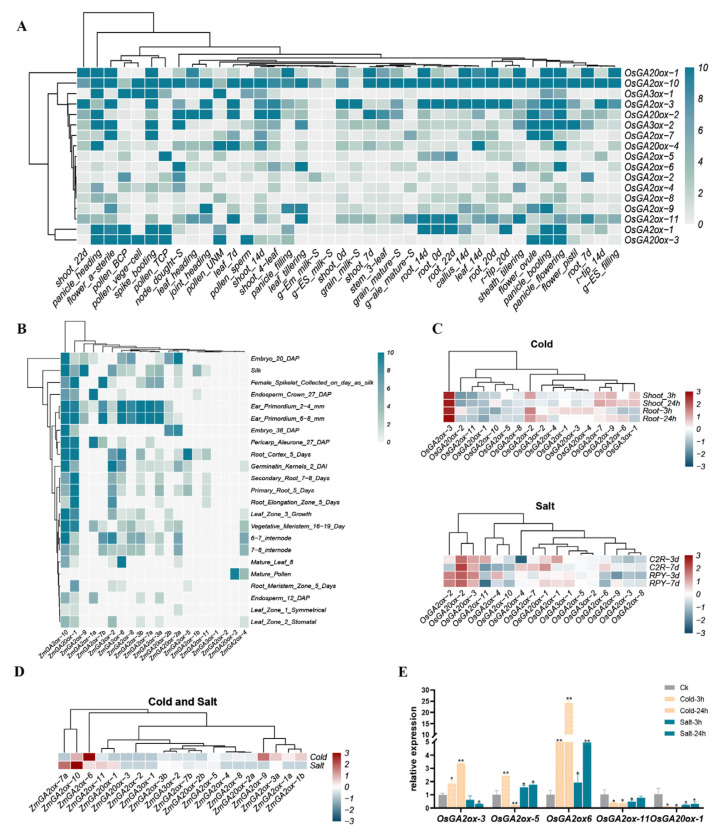
Analysis of the expression pattern of GAox genes. (**A**) Expression profiles of OsGAox genes in various tissues of rice. (**B**) Expression profiles of ZmGAox genes in various tissues of maize. (**C**) Expression profiles of GAox genes in rice under cold and salt stress. (**D**) Expression profiles of GAox genes in maize under cold and salt stress. (**E**) Relative expression profiles of five relative expression profiles in the roots of japonica rice (Nipponbare) treated with cold or salt for 3 h and 24 h.

**Table 1 genes-13-00863-t001:** Ka/Ks values and the divergence times of duplicate gene pairs in six tested species.

Seq-1	Seq-2	Ka	Ks	Ka_Ks	Duplication Type	Divergence Time (Mya)
*BdGA20ox-3a*	*BdGA20ox-3b*	0.03	0.08	0.37	TD	4.24
*ZmGA20ox-2a*	*ZmGA20ox-2b*	0.03	0.09	0.31	WGD/SD	5.04
*ZmGA2ox-3a*	*ZmGA2ox-3b*	0.04	0.10	0.45	WGD/SD	5.40
*ZmGA2ox-7a*	*ZmGA2ox-7b*	0.04	0.10	0.39	WGD/SD	5.43
*HvGA2ox-4*	*HvGA2ox-3*	0.06	0.21	0.31	WGD/SD	11.38
*SiGA2ox-11a*	*SiGA2ox-11b*	0.20	0.25	0.83	TD	13.53
*BdGA2ox-8a*	*BdGA2ox-8b*	0.14	0.35	0.40	TD	19.12
*OsGA20ox-2*	*OsGA20ox-4*	0.22	0.36	0.61	WGD/SD	19.82
*SiGA3ox-1*	*SiGA3ox-2*	0.32	0.37	0.87	WGD/SD	20.42
*SbGA2ox-3*	*SbGA2ox-4*	0.18	0.41	0.44	WGD/SD	22.47
*ZmGA2ox-3a*	*ZmGA2ox-4*	0.19	0.44	0.43	WGD/SD	24.40
*SiGA2ox-9*	*SiGA2ox-6*	0.19	0.45	0.42	WGD/SD	24.52
*BdGA2ox-7*	*BdGA2ox-10*	0.20	0.47	0.43	WGD/SD	25.73
*SiGA2ox-8*	*SiGA2ox-3*	0.33	0.47	0.71	WGD/SD	25.83
*HvGA2ox-8a*	*HvGA2ox-8b*	0.20	0.49	0.42	TD	26.84
*SbGA3ox-2*	*SbGA3ox-1*	0.34	0.51	0.68	WGD/SD	27.76
*BdGA20ox-4*	*BdGA20ox-2*	0.26	0.51	0.50	WGD/SD	27.86
*OsGA3ox-2*	*OsGA3ox-1*	0.33	0.51	0.64	WGD/SD	28.00
*ZmGA3ox-2*	*ZmGA3ox-1*	0.33	0.53	0.63	WGD/SD	28.87
*OsGA2ox-3*	*OsGA2ox-4*	0.26	0.53	0.49	WGD/SD	29.37
*ZmGA2ox-6*	*ZmGA2ox-9*	0.20	0.54	0.38	WGD/SD	29.43
*OsGA2ox-9*	*OsGA2ox-6*	0.21	0.54	0.39	WGD/SD	29.45
*SiGA2ox-7*	*SiGA2ox-3*	0.36	0.54	0.67	WGD/SD	29.55
*SbGA20ox-2*	*SbGA20ox-4*	0.27	0.55	0.50	WGD/SD	30.28
*BdGA2ox-9*	*BdGA2ox-6*	0.20	0.58	0.34	WGD/SD	32.10
*OsGA2ox-3*	*OsGA2ox-8*	0.31	0.59	0.53	WGD/SD	32.26
*SbGA2ox-9*	*SbGA2ox-6*	0.21	0.59	0.36	WGD/SD	32.35
*BdGA2ox-7*	*BdGA2ox-3*	0.41	0.61	0.67	WGD/SD	33.41
*BdGA2ox-2*	*BdGA2ox-1*	0.24	0.71	0.34	WGD/SD	39.18
*SbGA2ox-2*	*SbGA2ox-1*	0.23	0.78	0.29	WGD/SD	42.60
*OsGA2ox-2*	*OsGA2ox-1*	0.49	0.99	0.50	WGD/SD	54.19
*OsGA2ox-7*	*OsGA2ox-10*	0.46	1.06	0.43	WGD/SD	58.07
*SiGA2ox-2*	*SiGA2ox-1*	0.23	1.21	0.19	WGD/SD	66.43

## Data Availability

All data generated or analyzed during this study are included in this published article.

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
