# Peer review of "Genome-Wide Identification and Evolution Analysis of the Gibberellin Oxidase Gene Family in Six Gramineae Crops"

_genes, 2022, doi:10.3390/genes13050863_

Round 1

Reviewer 1 Report

Congratulation for the interesting research. Some small correction of the manuscript will be proposed below.

Introduction

Lines 71-73: please reformulate the sentence. Application of a gene has no meaning.

Material and methods

Line 121: Homologs analysis should be replaced with homology analysis 

Line 123: software [39]. ,an  - there is an extra punctuation mark, please delete

Results

Line 162: of which Zm was most and Si was the least in number of GAox proteins. - most and least are not the proper words in the sentence. Please reformulate. 

Line 186: GAox genes of six sepcieas were labeled by various colors. - please replace sepcieas with species

Line 204: GA2ox, GA3ox and GA20ox subfamilies - and should be written with no Italics

Line 208: in Os, HV, Zm, Si, Sb - please correct HV with Hv

Line 214: all gene pairs were subjected purification selection - please add subjected to purification selection

Line 224: sativa ssp. Japonica (A) - please write with small caps japonica

Lines 239-240: motif11, motif14, motif15, motif 7 - please use uniformly with or without space 

Discussions

Line 351: Liu et al conducted - please correct Liu et al. 

The following article should be on your interest.

Hu, L.; Wang, P.; Hao, Z.; Lu, Y.; Xue, G.; Cao, Z.; Qu, H.; Cheng, T.; Shi, J.; Chen, J. Gibberellin Oxidase Gene Family in L. chinense: Genome-Wide Identification and Gene Expression Analysis. Int. J. Mol. Sci. 202122, 7167. https://doi.org/10.3390/ijms22137167

Author Response

Thanks for your valuable suggestions! I have revised all your suggested revisions one by one. And the references you provided are useful for my research, I inserted it in line 27. The revised manuscript is uploaded in the attachment

Reviewer 2 Report

Manuscript entitles “Genome-wide Identification ………….. Six Gramineae Crops” deals with the comparative genomic 18 study of GAox genes in six Gramineae representative crops. Authors reported that GAox genes were predominantly expressed in the panicles of rice and maize. Authors also highlighted role of GAox genes in abiotic stress.

Study is good and need following clarification/suggestions

  1. What is basis for the selection of these six crops? Why not other Gramineae crops?
  2. Why study is focused on Gramineae only?
  3. Bo wet-lab experiment? At least expression analysis of representative GAox gene of each crop may performed under stress conditions.
  4. Only base on In silico analysis, their role in stress conditions were demonstrated. This is just a concept, to prove it a wet lab experiment is needed.
  5. If possible, provide genome organization of GAox genes.
  6. Line 20: ssp.?

Author Response

    Thank you very much for your advice! I have tried my best to complete the revision of the manuscript within the specified deadline, and the result is as follows:

  1. We chose these six cereal crops because their high quality genomes have already been reported. The genomes of other cereal crops, such as wheat, were not selected because they are tetraploid and therefore not suitable for further comparative analysis
  2. Because grass crops have important economic value, and GAox genes is widely used in rice and wheat breeding and is related to domestication, the identification of homologous genes of GAox genes can provide a reference for breeders

  3. I have performed qRT-PCR to verify several GAox that were significantly induced by stress in the latest manuscript.
  4. I do agree with your suggestion. It is indeed necessary to further verify our assumptions, but given the limited time, we will further verify through transgenic experiments in the future.
  5. I'm sorry I don't know much about the analysis of genome organization of genes, but I  provided the gene structure in the manuscript(figure 3B). I will strengthen my knowledge in this area!
  6. We have revised.

    Thanks again for your valuable suggestions for the manuscript!

Reviewer 3 Report

It is a well-formatted manuscript according to the trend of recent genome wide analysis papers. However, over time you may find that you need to show more and more information. In this paper, it is necessary to check gene expression for specific genes through qRT-PCR. Additionally, data showing that it is a complement to Arabidopsis or a VIGS-like method that can be quickly applied must be added.

It is necessary to introduce as an example the contents of lines 382-383, that applying the GAox gene is linked to plant stress resistance.

In Figure 5, the data should be structured so that the data on gene expression for salt and cold can be compared and analyzed in rice and maize. If not, it seems appropriate to make a comparison with other stresses.

It is recommended to insert a figure containing information about the intracellular location of genes. If everything is a prediction, this paper should be viewed as a review paper. The part that introduces direct data is more required.

Author Response

    Thank you for your valuable advice!I tried my best to complete the revision of the manuscript within the specified time limit. The results are as follows:

  1. We do agree with your suggestion. It is necessary to further test our hypothesis, but given the limited time, we hope to further validate this hypothesis in the future through full-scale transgenic experiments. In addition, I have performed qRT-PCR to verify several GAox that were significantly induced by stress in the latest manuscript.
  2. We have made changes based on your comments. In the latest manuscript of line420-422.
  3. Thank you for your suggestion, the data structure in Figure5 is really unreasonable, we have corrected it in the latest manuscript.
  4. Your suggestion is professional. We have considered the analysis of subcellular localization before, but found that some OsGAoxs genes have already been subjected to this experiment in some articles, and we cannot obtain germplasm resources of other species for the experiment in the short term. We therefore focused our analysis on the evolutionary pattern of GAox genes in graminaceous crops.

    Thank you again for your valuable advice!Best regards!

Round 2

Reviewer 3 Report

I know that it is difficult for authors to conduct experiments on revisions in a short period. However, if you add a little more data, it will be a helpful research paper for more researchers.